# Glucocorticoid receptor wields chromatin interactions to tune transcription for cytoskeleton stabilization in podocytes

Hong Wang[1,4], Aiping Duan[1,2,4], Jing Zhang[1], Qi Wang[1], Yuexian Xing[2], Zhaohui Qin [3], Zhihong Liu [1,2✉] & Jingping Yang [1,2✉]

Elucidating transcription mediated by the glucocorticoid receptor (GR) is crucial for understanding the role of glucocorticoids (GCs) in the treatment of diseases. Podocyte is a useful model for studying GR regulation because GCs are the primary medication for podocytopathy. In this study, we integrated data from transcriptome, transcription factor binding, histone modification, and genome topology. Our data reveals that the GR binds and activates selective regulatory elements in podocyte. The 3D interactome captured by HiChIP facilitates the identification of remote targets of GR. We found that GR in podocyte is enriched at transcriptional interaction hubs and super-enhancers. We further demonstrate that the target gene of the top GR-associated super-enhancer is indispensable to the effective functioning of GC in podocyte. Our findings provided insights into the mechanisms underlying the protective effect of GCs on podocyte, and demonstrate the importance of considering transcriptional interactions in order to fine-map regulatory networks of GR.

[1] Medical School of Nanjing University, Nanjing, Jiangsu, China. [2] National Clinical Research Center for Kidney Disease, Jinling Hospital, Medical School of Nanjing University, Nanjing, Jiangsu, China. [3] Department of Biostatistics and Bioinformatics, Rollins School of Public Health, Emory University, Atlanta, GA, USA. [4] These authors contributed equally: Hong Wang, Aiping Duan. ✉email: liuzhihong@nju.edu.cn; jpyang@nju.edu.cn

Glucocorticoids (GCs), such as the cortisol mimic dexamethasone (DEX), are the first line of treatment for various diseases. GCs bind to the glucocorticoid receptor (GR) and GR translocates to the nucleus to regulate transcription. Although GR is expressed ubiquitously, the GR has shown itself to be associated with cell identity. GR can maintain insulin secretion in β-cell[1] and mediate whitening of beige adipocyte[2].

Podocyte, with exquisite actin cytoskeleton to maintain its structure, is critical for normal kidney function. It adheres to the outer surface of the glomerulus to form a normal glomerular filtration barrier and is sensitive to various stimuli. Any damage to its cytoskeleton will disrupt the filtration barrier, lead to the massive leakage of protein into the urine, and eventuate kidney disease[3]. Previously, it has been assumed that GCs cure podocytopathy primarily by dampening the immune response. However, an increasing body of evidences indicates that GCs could attenuate podocytopathy by direct action on podocyte[4,5]—by enhancing actin cytoskeleton[6]. DEX treatment can recover disrupted podocyte cytoskeleton caused by LPS. However, with podocyte-specific deletion of GR, DEX cannot rescue the phenotype[7]. Additionally, GR in other kidney epithelial cells, like parietal epithelial cells, is not necessary for podocyte homeostasis[8]. Therefore, GR in podocyte directly effectuates protection by enhancing the cytoskeleton. However, how GR accomplishes this task has been understudied.

Upon translocation to the nucleus, GR either binds directly to genomic glucocorticoid response elements (GREs) or does so indirectly by tethering to other transcription factors (TFs) at already accessible regions[9]. It has been previously speculated that cell-selective GR binding tends to target pre-accessible chromatin determined by lineage-specific transcription factors[10]. However, functional effect measurement indicates that most responsive GR sites are not those pre-determined GR sites but those direct GR-binding sites[11,12]. The overall cell-type-specific distribution does not explain the cell-specific regulation of GR. As typical enhancers, GR elements could regulate target genes through chromatin loops[13]. The GR site at *GILZ* loci can targets and activates different promoters in different cell types[14], but how the genome-wide chromatin interactions contribute to cell-identity regulation is not yet understood.

Here, with podocyte as a model for studying the role of GR on cell-identity-related regulation, we captured and integrated transcriptome with GR binding, epigenetic state, and chromatin organization. We found that most GR bindings in podocyte possess the features of responsive GR sites[11]. Remarkably, the transcription-centered 3D interactions revealed that GR sites in podocyte were enriched at interaction hubs and super-enhancers (SEs). The enrichment at SEs was found to be conserved at responsive GR sites in other cell types, such as A549. Within the GR regulatory network in podocyte, we tested *ZBTB16*, a gene targeted by the top GR-activated SE, and confirmed that *ZBTB16* mediated the primary effect of GC on podocyte: the enhancement of cytoskeleton. Collectively, we uncover a strategy used by GR to regulate transcription through SEs and long-range interactions.

## Results

**GR casts cell-type-specific binding in podocyte.** In order to understand the role of GR in human podocyte (hPC), we treated hPC with 1 μM DEX, as the concentration is comparable to the in vivo therapeutic GC levels[4]. As previously reported, DEX treatment strengthened the cytoskeleton in non-dividing hPC (Supplementary Fig. 1). With the presence of DEX, we captured the reproducible genome-wide GR-binding profile of hPC (Supplementary Fig. 2a). The distribution is consistent with that in hPC treated with GC prednisolone[15]. The two profiles highly

overlapped (Supplementary Fig. 2b). At reported GR targets in podocyte, like *SAA1* and *SERPINE1*[16], the profiles displayed the same occupancy (Supplementary Fig. 2c).

To examine the cell-type-specific features of GR occupancy in hPC, we compared the GR distribution in hPC to its distribution in other human cell types including MCF7[17], K562[18], A549[19], Hela[20], and BEAS-2B[21]. These comparisons revealed both shared and unshared binding sites (Fig. 1a and Supplementary Data 1). The *PER1* region, which showed DEX response in multiple cell types[22], harbored a GR-binding site that is common to different cell types. In contrast, at loci like *Chr1:214355kb-214375kb*, *PKIA*, and *SCN9A*, the occupancy of GR was cell-type-specific. In genome-wide, 16.9–78.6% of GR-binding sites in hPC are shared (Fig. 1b and Supplementary Fig. 3). 21.4–83.1% of the GR-binding sites are unique to podocyte.

As it has been suggested that the cell specificity of GR sites is due to pre-determined accessibility by lineage-specific transcription factors[9,10,22], we continued to investigate the determinant of GR binding in hPC. We performed motif analysis on groups of GR sites (Supplementary Data 2) to understand the preference of GR binding. For MCF7-specific GR sites, the most enriched motifs were those of the forkhead family, which occupied 59.6% of the total sites. Motifs for the GATA family were enriched in K562-specific sites. GATA motif was found in 66.85% of those sites (Fig. 1c). Motifs for AP-1 complex as Fra1/2 were also enriched in these cell types. FOXA1 and GATA3 have been shown to be pioneer factors in MCF7 and K562, as they are the primary factor to bind DNA and modify the chromatin[18,23]. However, we found that the GRE motif is the most enriched motif in GR-binding sites in hPC (Fig. 1c). 93.3% of shared hPC sites and 63.3% of unique hPC possessed the sequences for GRE. This suggests that, for DEX-induced GR binding in hPC, GR likely binds directly to GRE, rather than being indirectly recruited by other factors or binding following other pioneer factors. In assessing our data, we concluded that GRE-driven GR bindings are also cell-type-specific.

**hPC GR-binding sites display features of responsive GR sites.** Compared to GR binding around other motifs, direct GR binding at GRE potentiated GC-induced enhancers[11]. Since most GR occupancy in hPC involves GRE, we hypothesize that most of these GR sites in hPC function as active regulatory elements. By examining the genomic location of the GR sites, we found 4.6% of these GR sites located at promoters (Fig. 2a). Among them, a large part (95.4%) of the hPC GR sites presented as non-promoter elements. Additionally, around 80% of the GR sites were located more than 5 kb away from TSS (Fig. 2b), suggesting that most GR sites are remote regulatory elements.

We then examined whether those potential regulatory elements would be induced after DEX treatment. To do this, we checked the level of H3K27ac which is an indicator of the activation of regulatory elements (Supplementary Fig. 4a). At sites with GR binding in hPC—either shared or unique—we found that there was a prominent increase in H3K27ac levels after exposure to DEX. In contrast, at sites without GR occupancy in hPC, there was no change in H3K27ac levels (Fig. 2c and Supplementary Fig. 4b). These results hold true genome-wide. At sites where GR binds in hPC, the level of H3K27ac increased. However, at sites where GR bound in other cell types but not in hPC, the occupancy of H3K27ac did not increase after exposure to DEX (Fig. 2d). This activation of GR sites is consistent across different genomic categories (Supplementary Fig. 5). These results indicate that GR in hPC binds to selective proximal and distal regulatory elements and activates these elements after DEX treatment, like a previously described group of "responsive GR sites"[11].

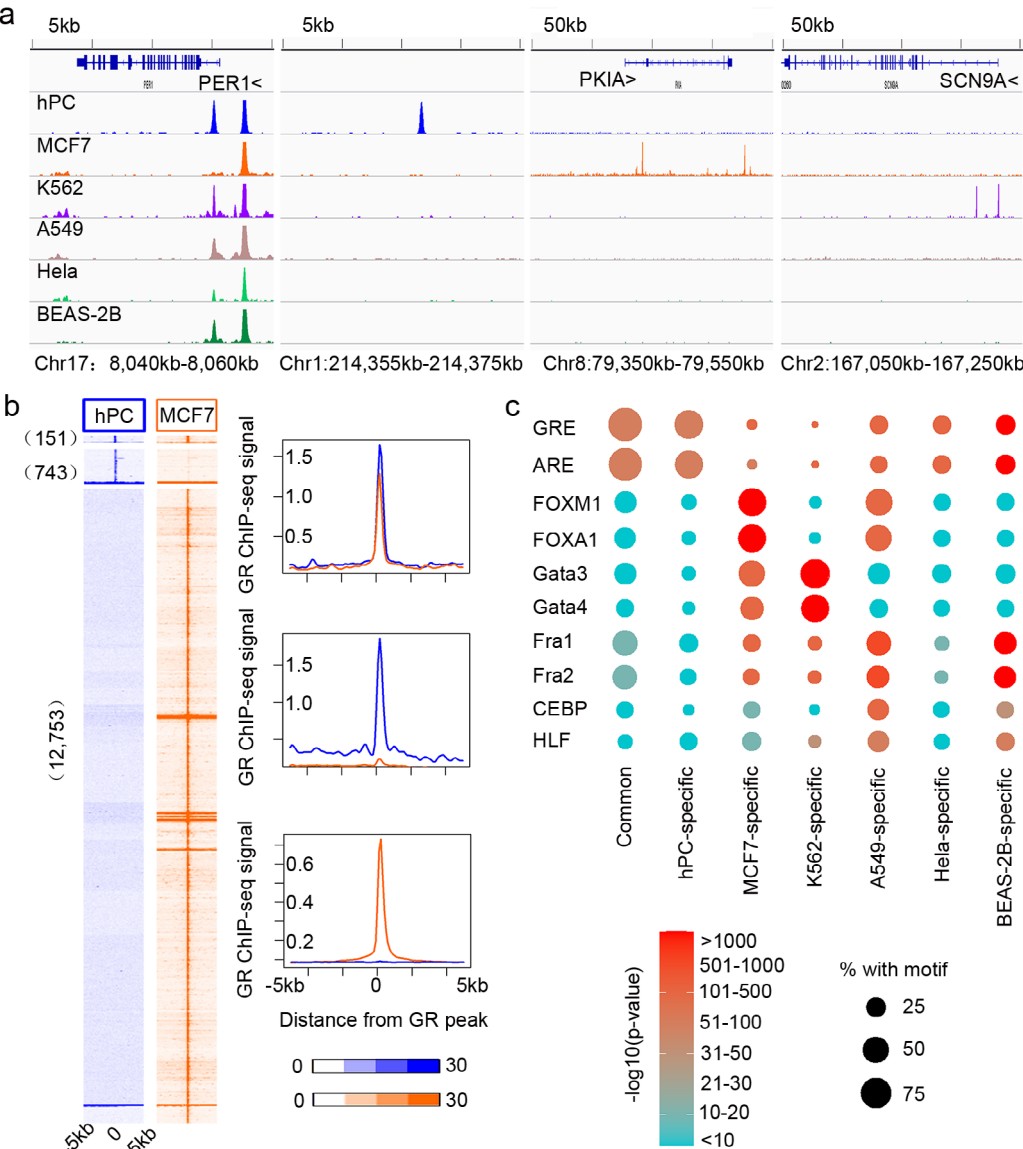

**Fig. 1 Characterization of GR binding in hPC. a** Genome browser view for GR occupancy in different cell types at *PER1*, Chr1:214355kb-214375kb, *PKIA*, and *SCN9A*. **b** Heatmaps and average profile of GR ChIP-seq signals at shared, hPC-specific, MCF7-specific sites. **c** Motifs analysis for different groups of GR sites. Color bar indicates −log10 (*p*-value), and the size of the circle indicates percentage with a motif.

**Transcription-centered chromatin interactions reveal responsive GR sites enriched at SEs.** As GR sites in hPC were inducible regulatory elements, we then investigated their effect on cell-identity-related transcription. The resulted transcriptome after DEX were related to the cytoskeleton (Supplementary Fig. 6a), but the change of transcriptome was loosely correlated with the activation of GR elements as exemplified with *BIN1* or *IRS2* loci (Supplementary Fig. 6b). The potency of GR for nearest gene activation was further reduced with the distance (Supplementary Fig. 6c). As chromatin interactions have been proven necessary for transcription regulation of distant enhancers, we then generated a transcription-centered chromatin interaction profile in hPC through HiChIP with H3K27ac to understand transcription regulation (Supplementary Fig. 7a). Nearly all the interactions embraced H3K27ac at one or both anchors (Fig. 3a). There were around 70% interactions between enhancers (E–E), about 2% between promoters (P–P), and about 20% were promoter–enhancer interactions (E–P) (Fig. 3b). For genes without GR-binding nearby, like *IRS2*, the transcription was upregulated as its promoter

interacted with a GR-activated enhancer more than 200 kb away, as well as with several activated elements in between (Fig. 3c). And yet, although there was a GR site within the *BIN1* gene body, the expression was not changed as its promoter did not interact with the GR site (Fig. 3c). Throughout the genome, 287 loops were strengthened and 260 loops were weakened after DEX (Supplementary Fig. 7b). The changes in 3D interactions are a combinatory result with changes of H3K27ac signal at the anchors, as well as true physical interaction dynamics. Either way, this would affect the regulatory potential. The percentage of upregulated genes is three times higher among genes associated with strengthened loops than others, while the percentage of downregulated genes is 1.8 times higher among genes associated with weakened loops (Supplementary Fig. 7c), confirming that the interactions were transcription-centered.

We continued to examine the features of GR-involved interactions. As expected, GR-associated interactions were more enriched in strengthened interactions than weakened or unchanged interactions (Supplementary Fig. 8a). The sizes of

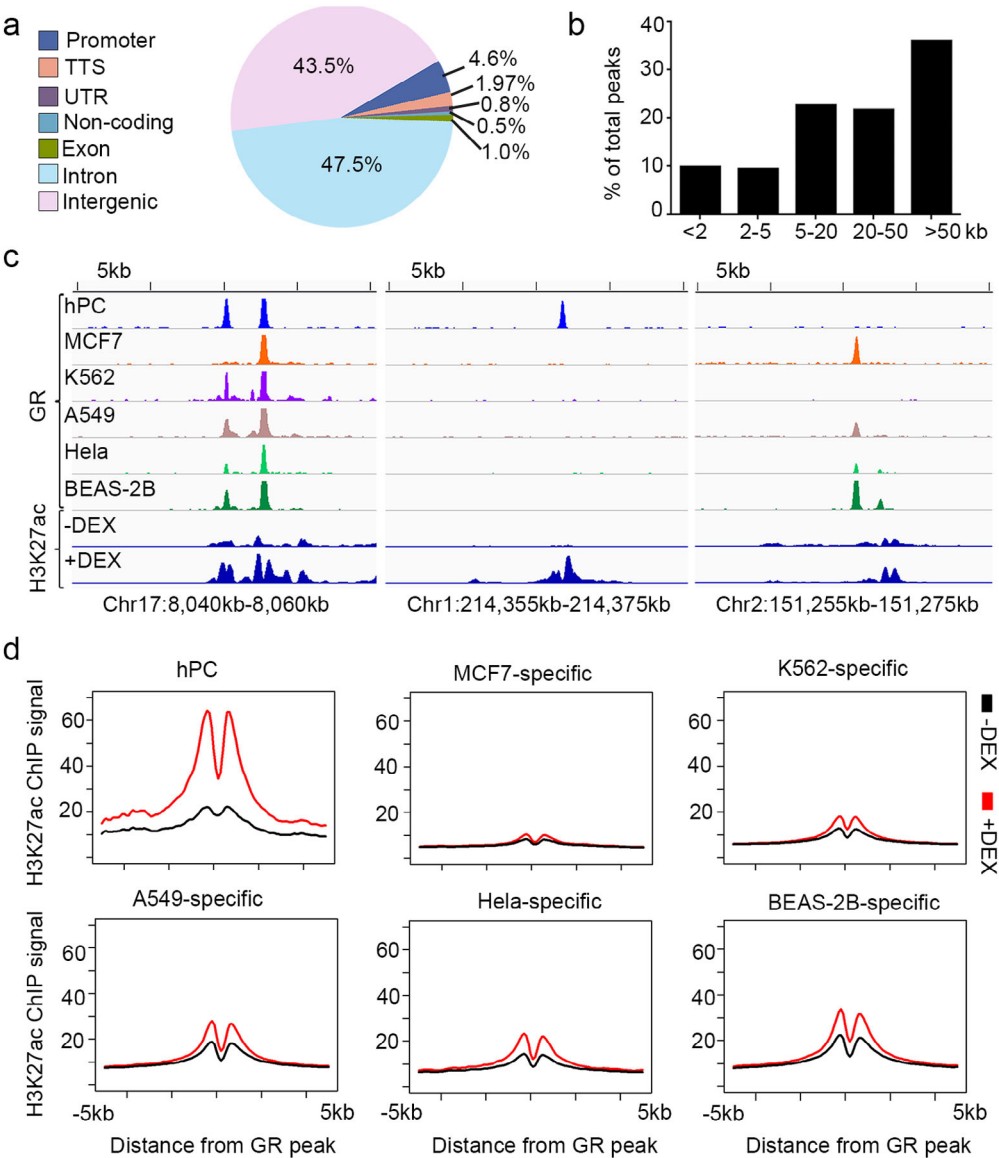

**Fig. 2 Classification and activity quantification of GR elements in hPC. a** Pie chart shows the distribution of GR-binding sites in hPC based on genomic annotation. **b** Histogram shows the distribution of distance between GR sites and nearest TSS. **c** Genome browser view of H3K27ac signals at loci with hPC shared (left), hPC unique (middle), or non-hPC (right) GR-binding sites. The top six tracks show GR signals in different cell types, and the bottom two tracks show H3K27ac in hPC with or without DEX treatment. **d** Average signals plot of H3K27ac in hPC with or without DEX stimulation at hPC, MCF7-specific, K562-specific, A549-specific, Hela-specific, or BEAS-2B-specific GR sites.

interactions were not different between GR and non-GR interactions (Supplementary Fig. 8b and Supplementary Data 4). However, the number of HiChIP interactions involving GR anchors was higher than non-GR anchors (Fig. 4a), suggesting that GR sites might act as interaction hubs. It has been suggested that SEs could serve as interaction hubs and that interaction hubs are responsible for SE function[24,25]. When we examined the ratio of SEs (Supplementary Data 3) to typical enhancers, the results showed that GR sites in hPC associated with SEs ten times more than they did in genome-wide (Chi-square test) (Fig. 4b). SEs with GR also showed increased H3K27ac after DEX treatment (Fig. 4c). There are various characteristics shared between GR in hPC and the responsive GR sites described in A549[11], such as enrichment for GRE motif and increased H3K27ac activity. Considering this, we went further to check if these above features of GR were shared by responsive GR sites in A549. Unlike HiChIP interactions, Hi-C interactions generated in A549 did not

evince differences between responsive GR sites and non-responsive GR sites (Fig. 4d), probably because Hi-C interactions also included non-transcription interactions. However, SE enrichment is conserved for responsive GR sites in A549. The enrichment was much higher there than genome-wide and also significantly higher than non-responsive GR sites (Fig. 4e). The responsive GR sites that associated SE also showed increased H3K27ac after DEX, while SEs with non-responsive GR did not show significant shifts (Fig. 4f).

It is well known that SEs control the genes for cell identity[26], so we checked the target genes of GR-associated SEs in hPC and Res-GR-associated SEs in A549. We identified the potential target genes through chromatin interactions. The gene ontology analysis of these genes indicated genes that interacted with GR-associated SEs were related to hPC identity as cell adhesion (Fig. 4g and Supplementary Data 5), and genes that interacted with Res-GR-associated SEs were involved in lipid metabolism and cancer pathway in A549 (Fig. 4h).

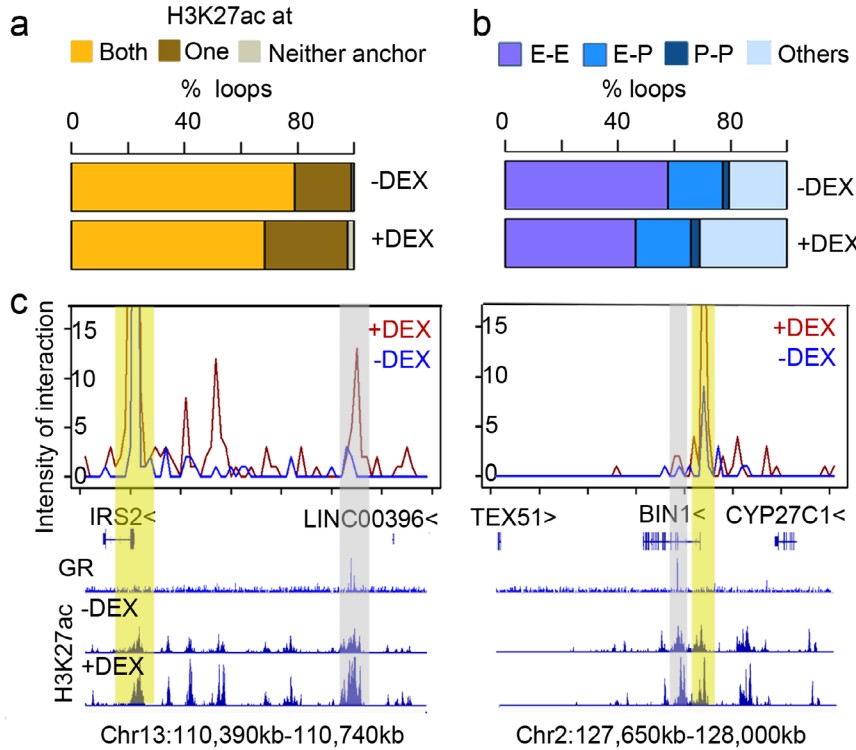

**Fig. 3 Transcription-centered interaction loops captured by H3K27ac Hi-ChIP. a** The stack plot shows the proportion of H3K27ac within both, one, or neither anchors. **b** The proportion of E–E (enhancer–enhancer), E–P (enhancer–promoter), P–P (promoter–promoter) interactions in HiChIP profile. **c** Virtual 4C profile at IRS2 or BIN1 locus with GR, H3K27ac signal with or without DEX treatment. The yellow bar highlights the promoter of IRS2 or BIN1. Gray bar highlights the GR sites.

Taken together, the high proportion of responsive GR sites in hPC and transcription-centered chromatin interactions revealed responsive GR sites were enriched at SEs and could mediate cell-specific- and identity-related-effects in GC response.

**Target gene of the top GR-associated SE mediates maintenance of hPC cytoskeleton.** The levels of H3K27ac at GR-associated SEs could increase up to 45 times (Supplementary Fig. 9a). The Top1 SE (SE1) remotely targeted the promoter of *ZBTB16* (Fig. 5a). We confirmed that *ZBTB16* was expressed in podocyte of the human kidney (Fig. 5b). Furthermore, under podocythopathy, expression of *ZBTB16* in the kidney (data from Nephroseq https://www. nephroseq.org) (Fig. 5c and Supplementary Data 6), which was confirmed to occur in podocyte through single-cell RNA-seq during our preparation of this manuscript, is greatly downregulated[27]. Although the GR binding at SE1 was not hPC-specific (Supplementary Fig. S9b), this GR binding was functional in hPC indicated by H3K27ac. After exposure to DEX, SE1 was activated in hPC and the activity of the *ZBTB16* promoter was highly enhanced as indicated by the H3K27ac level and RNA-seq signal (Fig. 5a).

To check the role of ZBTB16 in GCs protection of podocyte, we knocked down the expression of *ZBTB16* by siRNAs and examine the primary effect of GC on podocyte: the enhancement of cytoskeleton. The cytoskeleton of hPC was disorganized after treating with *ZBTB16* siRNA (Fig. 5d). The quantification of F-actin showed decreased actin filaments (Fig. 5e and Supplementary Data 7). When we examined the expression of marker genes for hPC cytoskeleton, we found *SYNPO* was significantly upregulated by DEX but without any GR binding involving *SYNPO* (Fig. 5f and Supplementary Fig. S9c). After knocking down *ZBTB16*, the increase of *SYNPO* expression after DEX was significantly inhibited (Fig. 5g and Supplementary Data 8),

confirming the necessity of ZBTB16 for GR functionality on *SYNPO*. ZBTB16, with a POZ-Krüppel (POK) domain that can bind to specific DNA sequences, has been reported to be a transcriptional regulator[28]. ChIP-qPCR confirmed that *ZBTB16* bound to the promoter of *SYNPO* (Fig. 5h and Supplementary Data 8). Thus, through chromatin interaction, GR-associated SEs targeted and activated promoters of other transcription factors like *ZBTB16*, which in turn extended transcription regulation to more effector genes.

Altogether, our results indicate that GRE-driven GR binding can be cell-type-specific and that it activates selective regulatory elements, especially SEs. Via chromatin interactions, GR-associated SEs were wired to target promoters and cast cell-identity-related regulation on transcription networks (Fig. 5i).

## Discussion

GR is a transcription regulator of cellular responses to GCs in many cell types. Here, we integrated multi-layered epigenomic features to investigate its role on cell identity. The results revealed a unique characteristic of GR binding and its role in transcriptional regulation through SEs.

In our study, the cells were treated with DEX for a longer-term than they have been in previous studies. The GR motif was specifically enriched at GR sites in this study, indicating motif-directed GR binding under the longer-term inducement of DEX. This is consistent with the proposition that long-lasting GR peaks have stronger GR motif than transient-GR binding[12]. It is possible that the transient-GR binding is a byproduct with no function. In A549 treated for a short term, only a small fraction of GR sites showed an inducible signal of H3K27ac[11]. In contrast, a large proportion of long-term GR sites in this study showed increases of H3K27ac. These observations suggest that long-lasting GR sites would be more responsive than transient-GR

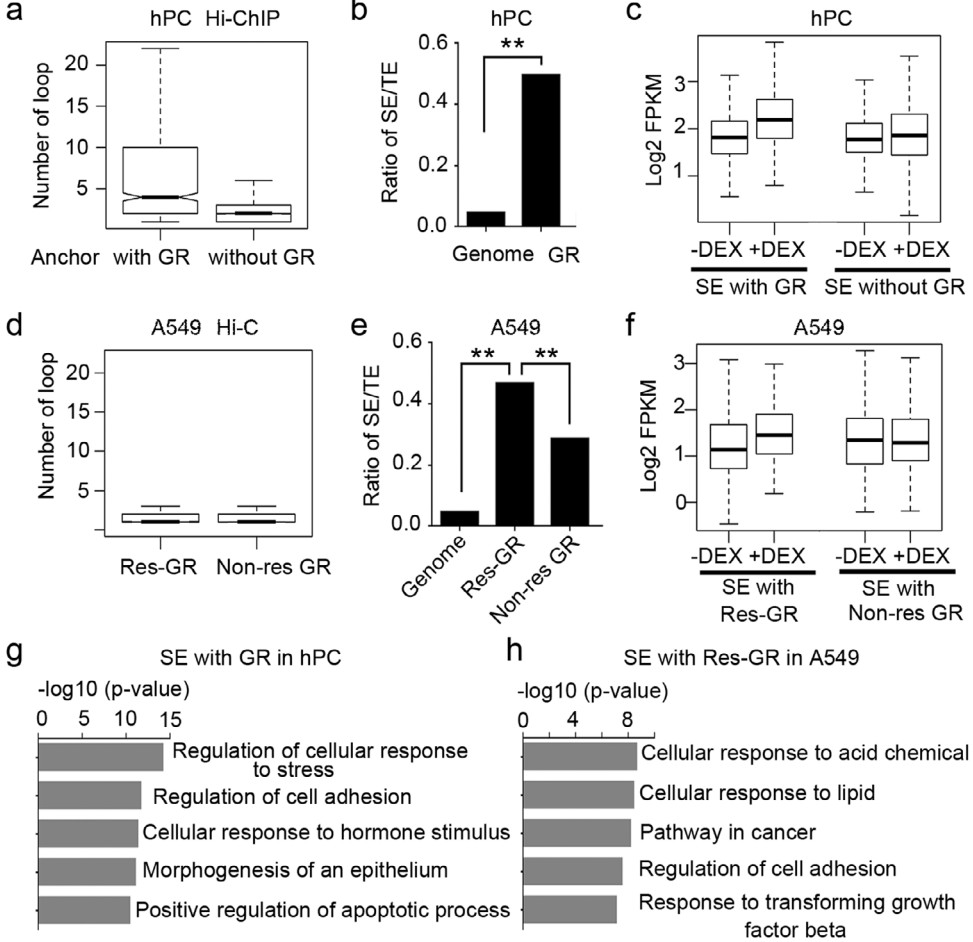

**Fig. 4 GR sites are enriched at SEs. a** Number of HiChIP loops for anchor with or without GR in hPC. Center line, median; box limits, upper and lower quantiles. **b** Ratio of SE (super-enhancer) over TE (typical enhancer) in the genome ($N_{SE} = 1409$ and $N_{TE} = 39,420$) and GR-binding regions ($N_{SE} = 242$ and $N_{TE} = 506$) in hPC, $p = 0$ (chi-square test, **$P < 0.01$). **c** Box plot of H3K27ac signal on SEs with or without GR binding in hPC. Center line, median; box limits, upper and lower quantiles. **d** Number of Hi-C loops for anchor with or without GR in A549. Center line, median; box limits, upper and lower quantiles. **e** Ratio of SE (super-enhancer) over TE (typical enhancer) in genome ($N_{SE} = 1189$ and $N_{TE} = 24,873$), responsive GR (res-GR, $N_{SE} = 390$ and $N_{TE} = 724$), and non-responsive GR (non-res-GR, $N_{SE} = 1090$ and $N_{TE} = 3719$) regions in A549, $p_{Res} = 0$, $p_{Nonres} = 0$ (chi-square test, **$p < 0.01$). **f** Box plot of H3K27ac signal on SEs with Res or non-res-GR binding in A549. Center line, median; box limits, upper and lower quantiles. **g**, **h** Top GO terms for the genes interacted with GR-associated SEs in hPC or Res-GR-associated SE in A549.

binding, though further in vivo study is necessary to validate this hypothesis.

Interestingly, though the GR site's function as an enhancer has been demonstrated in other cell types[12], the preference of GR sites for SEs has not been discovered. This is probably due to two reasons. First, it is easier to detect the relationship between SEs and GR at responsive GR sites, because SEs are particularly concentrated at responsive GR: most of the GR sites in this study present as responsive, whereas non-responsive GR sites are dominant in other studies. Second, because the chromatin interactions we captured through HiChIP with H3K27ac are transcription-centered, this study is more sensitive to features related to transcription. For example, there are more interactions involving responsive GR sites/SEs than other genomic regions. In contrast, Hi-C data in previous studyies[13] does not show a different number of interactions at responsive GR sites and enhancers without GR (Fig. 4d). Thus, transcription-centered chromatin organization would be more sensitive to and more useful for analyzing transcription regulation.

In addition to direct regulation by GR-associated SEs, there are also secondary effects that are mediated by GR-targeted TFs. For example, GR target gene *KLF15* facilitates GR function in lung epithelial cells, while target gene *KLF9* antagonizes GR effects in macrophage[29,30]. In this study, we identified ZBTB16 as one of many GR-regulated TFs. It in turn regulates *SYNPO* for the cytoskeleton in the differentiated podocyte. This multi-layered transcriptional regulatome results in a complex and dynamic transcriptional response to GCs. This could be part of the reason that GR can, depending on the context, be either anti-inflammatory or pro-inflammatory[31]. Through the complex regulatory network, GC treatment could remodel the transcription program of cell identity. It is reported that GR-regulated chromatin remodeling can be utilized by other factors such as FOXA1 or ER[17,32]. In this way, the long-lasting GR sites can transform the cell state through SEs and prepare the cells for responding to further treatment[33]. Our data suggest that GR binding on chromatin could enrich at SEs. Cancer cells are often addicted to the SEs driven transcriptional programs, and pharmacological targeting against SEs has shown great promise in reducing tumor growth and proliferation in several preclinical tumor models[34,35]. The use of GC might rewire the SEs profile in cancers and create a genomic environment with therapeutic effects. In lines with this hypothesis,

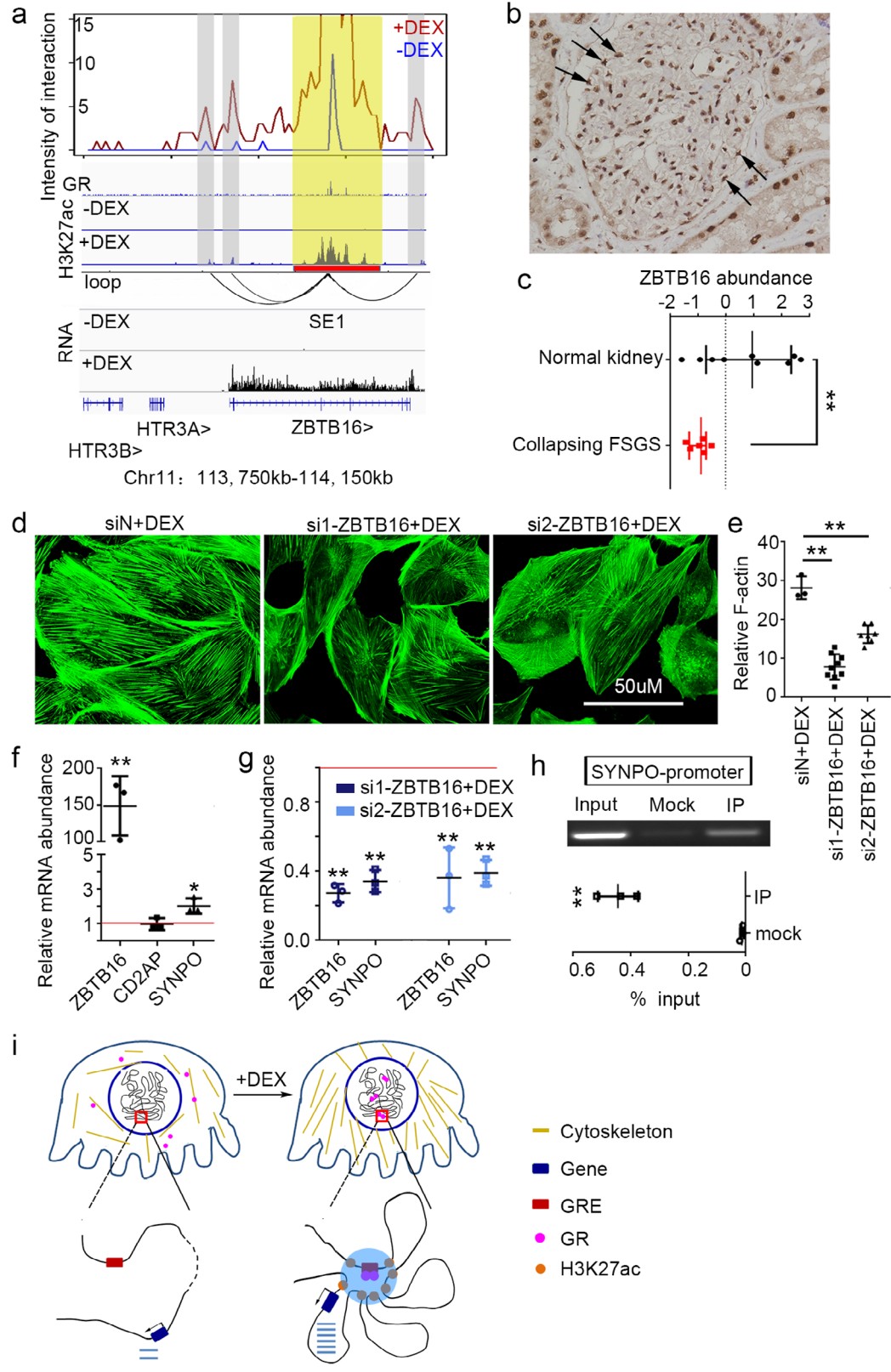

it has been demonstrated that adding GC to the therapy could improve ER+ breast cancer outcome[36], and that combination therapy with GC has synergistic effects for severe forms of lupus nephritis[37,38]. Further studies are needed to confirm the potential role of GR-associated SEs in cell-state transition and to develop better strategies for disease treatment.

## Methods

**Human podocyte culture and treatment**. Conditionally immortalized human podocytes were grown in RPMI-1640 medium with 10% FBS, 1% (v/v) insulin-transferrin-selenium (Gibco), and 1% penicillin–streptomycin at 33 °C in a 5% (v/v) $CO_2$ humidified incubator. Cells were differentiated at 37 °C. After being cultured at 37 °C for four days, podocytes were treated on the fifth day with or without 1 μM DEX for five additional days before harvest.

**Fig. 5 Role of Top1 SE in GR-mediated cytoskeleton stabilization. a** Virtual 4C profile at SE1 with GR signal, H3K27ac signal, RNA-seq signal, and detected HiChIP loops. Yellow bar highlights SE1 and anchor of virtual 4C. Gray bar highlights the interacted regions. **b** Immunostaining of ZBTB16 on human kidney specimen shows expression in podocyte. The arrows indicate part of podocytes in the glomeruli. **c** Expression of *ZBTB16* in glomeruli from Nephroseq for normal ($N = 9$) or diseased kidney as collapsing Focal Segmental Glomerulosclerosis (collapsing FSGS patients, $N = 6$). $p = 0.008$, * indicates $p < 0.05$, Center line, mean, upper, and lower standard deviation (SD). **d** Fluorescein-conjugated phalloidin staining for podocytes cytoskeleton with siRNAs against negative control (left) or *ZBTB16* (middle and right). Bars, 50 μM. **e** Quantifications of F-actin stress fibers as in **d**. $p_{si1-ZBTB16+DEX} = 0.0001$, $p_{si2-ZBTB16+DEX} = 0.0001$ (*$p < 0.05$, **$p < 0.01$) Center line, mean, upper, and lower standard deviation (SD). **f** Expression of *ZBTB16* and podocyte cytoskeleton genes in hPC without or with DEX treatment. $p_{ZBTB16} = 0.003$, $p_{CD2AP} = 0.951$, $p_{SYNPO} = 0.015$ (3 replications, *$p < 0.05$, **$p < 0.01$). Red line stands for relative expression at 1 as in control, Center line, mean, upper and lower standard deviation (SD). **g** Expression of *ZBTB16* and *SYNPO* under interference with siRNAs against *ZBTB16*, $p_{ZBTB16} = 0.0001$, $p_{SYNPO} = 0.0001$ for the group of si1-*ZBTB16* + DEX; $p_{ZBTB16} = 0.003$, $p_{SYNPO} = 0.0001$ for the group of si2-*ZBTB16* + DEX (3 replications, *$p < 0.05$, **$p < 0.01$). Red line stands for relative expression at 1 as in control, Center line, mean, upper and lower standard deviation (SD). **h** ChIP-qPCR of *ZBTB16* at *SYNPO* promoter, $p = 0.0004$ (3 replications, **$p < 0.01$), Center line, mean, upper, and lower standard deviation (SD). **i** Model depicts GR binding at super-enhancer and form interaction hubs to regulate genes in hPC-specific cytoskeleton stabilization.

**RNA-seq**. The total RNA was enriched by depletion of rRNA. The library was constructed and sequenced by Vazyme Biotech Company. Two biological replications were carried out for each condition.

**ChIP-seq and qPCR**. ChIP assays were performed according to the published protocol[39]. The chromatin was sheared on the Sonics VCX-130 with 15 s on, 30 s off, 12 cycle. Immunoprecipitation was performed using 3–5 μg of antibodies (GR, Santa Cruz Biotechnology, sc1003X; H3K27ac, Abcam, ab4729; ZBTB16, R&D Systems, MAB2911). DNA was purified by DNA Clean & Concentrator-5 kit (ZYMO). Libraries were constructed for ChIP samples on GR or H3K27ac with NEBNext Ultra DNA Library Prep Kit for Illumina (E7370), and sequenced on HiSeq4000 by Annoroad Gene Tech. (Beijing) Co., Ltd. Two biological replications were carried out for each condition. ChIP sample on ZBTB16 was used for qPCR with the primer at the promoters of *SYNPO* (forward: CATGAGTGGGGAAA CTGCAC; reverse: AGAGAGGTCTGAGGTTTGGC). Three biological replications were carried out.

**HiChIP**. HiChIP was performed according to the published protocol[40], with the following modifications: 10e6 cells were used for each sample; fixed and isolated nuclei were cut by *Nla*III restriction enzyme (NEB, R0125S); the nuclei with in situ-generated contacts were shared on Sonics VCX-130 with 15 s on, 30 s off, 4 cycles; the antibody to H3K27ac (Abcam, ab4729) was used for ChIP; two biological replications were conducted for each treatment.

**Tissue staining**. Kidney tissues from patients' renal biopsy were fixed in 4% buffered paraformaldehyde for two days, embedded in paraffin, and processed for sectioning. For immunohistochemistry staining, paraffin-embedded sections were deparaffinized and rehydrated. The following antibody was used: ZBTB16 (R&D Systems, MAB2944).

**Transfection**. Podocytes were cultured in 6-well plates to differentiate for 4 days and then treated with DEX for 3 days. After that, cells were transfected by siRNA using the Lipofectamine RNAiMAX reagent (Invitrogen, CA) following the manufacture's instructions. The transfection reagents were withdrawn 6 h later. Cells were harvested for RNA extraction or F-actin staining at 48 h.

**F-actin cytoskeleton staining and stress fiber quantification**. Transfected cells were fixed with 4% paraformaldehyde at room temperature. After washing with PBS, cells were permeabilized with 0.1% Triton X-100. F-actin was stained by rhodamine-labeled phalloidin and the staining was photographed with a fluorescence microscope. The images were digitized such that the areas of rhodamine-stained F-actin fibers could be converted to black pixels; quantification followed, using ImageJ software (NIH, Bethesda, MD) according to the method described in the software. The mean F-actin content of each image and the total F-actin content each cell were calculated and expressed as arbitrary units.

**RT-PCR and quantitative PCR**. The total RNA was isolated using the MiniBEST Universal RNA Extraction Kit (TaKaRa Code No. 9767). RNA purity and content were determined by Nano drop2000. cDNA was synthesized with PrimeScript™ RT Master Mix (TaKaRa Code NO. RR036A). qPCR was performed using the ChamQ SYBR Color qPCR Master Mix (Vazyme Q411-02) and carried out on an ABI Prism 7900 (Applied Biosystems). All quantitative data from the real-time RT-PCR were normalized by 18S as an internal control and calculated using the ΔΔCt-method. The primers for 18S were: forward 5′-TTTCTCGATTCCGTGGGTGG;

reverse 5′-AGCATGCCAGAGTCTCGTTC. The primers for *ZBTB16* were: forward 5′-CCTCAGACGACAATGACACGG; reverse 5′-CTCGCTGGAATGCTT CGAGAT. The primers for *SYNPO* were: forward 5′-GGGTCCCTGTCATGCT ACTT; reverse 5′-CCCAGAACCTGCCATGAATG. The primers for *CD2AP* were: forward 5′-ATCAAACGGGAAAGGCATGG; reverse 5′-TTGGTTTGTGG ATGTGGCTG.

### Bioinformatics
*RNA-seq analysis*. The adaptors-trimmed and quality-filtered reads were mapped to hg38 using HISAT2 with default parameters. Transcript assembly was performed using Stringtie[41,42]. Expression level estimation was reported as fragments per kilobase of transcript sequence per million mapped fragments (FPKM). Differentially expressed genes were identified using DESeq2[43] with cutoff at $p < 0.05$.

*ChIP-seq analysis*. The adaptors-trimmed and quality-filtered reads were aligned to the hg19 and hg38 using Bowtie2[44] with default parameters and uniquely mapped reads were used for peak calling with MACS2[45]. -q is 0.01 for GR and is 0.05 for H3K27ac. The GR peaks from two biological replications were combined after the filtering of ENCODE blacklist for subsequent analyses. The GR ChIP-seq in other cell types were downloaded from MCF7 (SRR2176969), K562 (EBI Array Express, E-MTAB-2955), A549 (SRR5093186), Hela (SRR067992), and BEAS-2B (SRR8485261). The SRA files were converted to fastq files using "sra-toolkit". The fastq files underwent the exact processes as GR ChIP-seq in hPC to call peaks. A pairwise comparison of GR peaks in cell types was performed as peak overlapping. hPC-specific GR peaks were identified by subtracting peaks in all other cell types. The heatmaps of ChIP-seq results were generated by Seqplots (Bioconductor package)[46]. De novo motifs were identified using "findMotifsGenome.pl" function in Homer[47] with the parameter "–size given" for GR-binding regions.

*HiChIP analysis*. HiChIP paired-end reads from four experiments were separately aligned to hg19 genome using the HiC-Pro pipeline[48] with default parameters to assign reads to *Nla*III restriction fragments, filter for valid interactions, and generate binned interaction matrices. Then the output from HiC-Pro was combined for Hichipper to call loops[49]. We filtered the loops with FDR 0.001 and loop distance more than 20 kb. The loop counts for each experiment were retrieved from the matrices generated by HiC-Pro above using in-house R script. HiC-Pro filtered reads were used to generate heatmaps with Juicebox[50]. Virtual 4C data were extracted with the Juicebox tool "dump". The promoter–promoter, promoter–enhancer, enhancer–enhancer interactions were annotated on Hichipper called interactions. Promoters were defined as 1 kb around RefSeq TSS of protein coding. Enhancers were defined as H3K27ac peaks not overlapping with promoter regions. Differential loops were generated by exactTest from edge R package ($p$-value < 0.05). Strengthened loops were those with fold-change of interaction frequency >0, and weakened loops were those with fold-change <0. The arc of loops were drawn with Diffloop[51]

*Super-enhancer identification*. SEs were identified using the ROSE (Rank Ordering of Super Enhancers) with default parameters ($-s$ 12500 $-t$ 2500)[34,52]. The SEs were called based on the H3K27ac ChIP-seq data from hPC with or without DEX treatment independently. The final SE profile was then generated as the union regions of the two SE lists. The same method was applied to data in A549 under GSE91248[53]. The SE lists are provided in Supplementary Data 3.

**Statistics and reproducibility**. The following statistical tests were performed or otherwise described in bioinformatics analysis: two-tailed Student $t$-test (Fig. 5c, e–h), chi-square test (Fig. 4b, e). Sample sizes and biological replications were described in the figure legends.

**Reporting summary**. Further information on research design is available in the Nature Research Reporting Summary linked to this article.

## Data availability

RNA-seq, ChIP-seq, and HiChIP-seq data used in this study have been deposited in the GEO database under the accession codes GSE117888. The secure token for GSE117888 is "ijirooowdpcrdc". Other data supporting the findings of this study are available within the paper and the Supplementary Data files or from the corresponding author upon reasonable request.

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

## Acknowledgements

We thank Noah Rawlings for proofreading the revised manuscript. J.Y. and Z.L. are grateful to support from the National Natural Science Foundation of China (81500515), Natural Science Foundation of Jiangsu Province (BK20150591), Nanjing University, and Emory University Collaborative Research Grants (NE2019003).

## Author contributions

J.Y. and Z.L. conceived and directed the research; H.W. and A.D. performed research; J.Y., H.W., J.Z., Q.W., and Y.X. contributed to data analysis; J.Y., H.W., Z.Q., and Z.L. co-edited the manuscript with input from all authors.

## Competing interests

The authors declare no competing interests.
