## [Peer Review File · Communications Biology]

Reviewers' comments:

Reviewer #1 (Remarks to the Author):

In this study, integrated GR ChIP-seq, H3K27ac ChIP-seq and HiChIP studies were performed in human podocytes. Comparisons are made between the podocyte data and published GR ChIP-seq data sets in two other cell types. ZBTB16 is defined as a podocyte-specific GR target and is implicated as a downstream transcriptional effector of cytoskeletal reorganization in response to induction of GR signaling.

Although there are numerous reports on GR ChIP-seq and other aspects of GR signaling in other cell types, there are no genome-wide reports on GR signaling in podocytes. Thus, the data presented here, based on what appear to be well-performed studies, will be a useful contribution to the literature. The conclusions made in the work are generally valid, although a more scholarly review of the literature encompassing additional references and current models of GR action would help contextualize these studies. Indeed, the two comparator cell types used in the primary analysis are not clearly representative of GR signaling responses in many other highly studied cell types, in which GR binding sites based on the consensus sequence strongly overlap with GR binding peaks defined by ChIP. In that regard, although ZBT16 was not identified as a GR target in the two comparator cell types, it is well known as a target in other non-podocyte contexts. Thus, specific suggestions to improve the impact of the work include:

- 1) Please compare and contrast GR regulation in this cell type with several other GR ChIP-seq data sets that are potentially more representative, such as/including A549, HeLa and BEAS-2B cells. Please specifically include analysis of ZBTB16 in this work. In that regard, it would also be useful to determine whether GR interacts with the synpo locus, which would imply a GR-ZBTB16 feed forward loop.
- 2) "Super enhancers" are cell type specific. The criteria that was used to define super enhancers in this work are not clear in the text (there are just three sentences in the supp methods), nor is it clear that dex-induced H3K27ac levels should be used as a criteria for super enhancer. Please provide more details in the text on this approach and justify why using dex-induced H3K27ac levels can be used to define a super enhancer – this seems like it could be a tautology.
- 3) Whether changes in three-dimensional interactions as assayed by HiC or similar approaches represent 1) changes in the efficiency of cross-linking of loops due to the absence or presence of transcription factors or 2) whether they in fact represent dynamic changes in chromatin structure, is not really clear. Please consider this first alternative in discussing induced and "repressed loops"
- 4) In many places the language is difficult to follow and uses arcane terminology and sentence structure. Editorial review of the text is suggested.
- 5) For maximal utility to the community, data needs to be also mapped in hg38

Reviewer #2 (Remarks to the Author):

In this manuscript, Wang et al use several genome-wide approaches to investigate GR action in a podocyte cell line model. The authors conclude that GR binds directly to many tissue-specific enhancers and that the GR cistrome is particularly enriched at super-enhancers. The authors also validate some of their findings by confirming the functional role of some novel target genes. While the work uses several interesting genomic approaches, I found some concerns regarding the experimental design chosen by the authors which need to be either addressed or properly justified as they may put into question some of their claims.

Major comments

1- The experimental design. The authors chose to work with a conditional transformed cell line model, wherein culturing these cells under 37C leads to podocyte maturation. According to ref.11, the maturation process takes 12-13 days without dexamethasone, but interestingly the differentiation process can be accelerated by DEX. In this manuscript, the authors only cultured the cells for 4 days, and then added DEX as a treatment for an additional 5 days. May I ask why

the authors chose this protocol? My specific concerns are 1) the maturation process was not completed, therefore the authors were not working with podocytes before DEX treatment, and 2) DEX was likely involved in the maturation process, therefore the authors are not studying the effect of GCs on matured podocytes but rather the effect of DEX on podocyte differentiation. Have the authors found mature podocyte markers before adding DEX, such as synaptopodin or nephrin (Saleem et al JASN 2002)? In this regard, Fig. S1 shows a big difference in phenotype before and after DEX. If the authors follow the 10-14 day maturation protocol as in ref. 12 (no DEX at all), How the cells would compare in phenotype?

2- The long DEX treatment. Performing a 5-day DEX treatment and then analyze genomic binding, epigenetic marks, and transcription can be tricky, as secondary effects are basically unavoidable (as the authors showed with the ZBTB16 and SYNPO genes). May I ask for the rationale for the longer treatment? What is the cell cycling duration on these cells? Do they grow during the 5-day treatment? I find hard to interpret the genomic data and transcriptional activity as any observed effect is likely due to secondary and even tertiary effects.

3- The experimental replicates. There is only one GR ChIP replicate which potentially puts these findings on somewhat shaky grounds. Even though the authors stated that the GR profile is consistent with another hPC study (ref. 12), the experimental design is very different (i.e. hPC matured for 10-14 days before treatments, use of Prednisolone 1 h instead of Dex for 5 days). The same replicate issue applies to the H3k27ac ChIPs (only one replicate).

4- ChIP comparisons. The authors compared their GR ChIP dataset with previously published data in hPC and other cell lines. Could the relatively lower number of peaks detected (~1000 peaks) compared to the other cell lines be related to the very long DEX treatment? Once again, I find the comparisons difficult as the experimental designs are so different.

5- Fig. 2. How comparable is the adult-kidney epigenome to mature podocytes? I am wondering how valid is to use the kidney datasets to segment the genome for the podocytes.

6- It is generally accepted that GR does not massively change the genome topology (Ref. 9). I wonder whether the effects seen in the HiChIP H3K27 experiment are due to the experimental design chosen by the authors. In other words, would the authors have observed similar changes with a shorter DEX treatment, or the changes observed are a consequence of the differentiation process during the 5-day DEX treatment?

Minor comments

1- Although the paper is readable, I strongly suggest the authors edit the manuscript with a native English speaker, as many grammar mistakes are present throughout the text.

2- Fig. S2. Please add the total number of peaks shown in the heat maps. Also, in the legend and the figure, McCaffey should be McCaffrey.

3- Fig. 1c and text. The authors compared motif enrichment between their dataset and others and concluded that "GR likely binds directly to GRE, rather than being indirectly recruited by other factors"; While I agree with the conclusion, the last paragraph in page 4 seems to imply that the presence of other motifs demonstrates indirect binding. This is not necessarily the case as the pioneer model (ref. 17) states that pioneer factors open the chromatin to allow the secondary factor (in this case GR) to bind directly to their response element. Perhaps the authors should rephrase that paragraph.

4- Fig. S4 is not very informative as it stands. I suggest the authors invest more space to explain in more detail the chromatin segmentation technique.

Reviewer #3 (Remarks to the Author):

- The need to study GR specifically in podocytes is not properly discussed in the Introduction
- Define "responsive GR site" versus "non-responsive GR site"
- Not clear why the authors are comparing hPC treated with DEX vs GC prednisolone (Fig. S2)
- Why are authors looking at BIN1 and IRS2 in Fig S6B?
- Define anchor (Fig. 3a). Explain why there is no difference +/-DEX in Fig 3a+3b
- In Fig.S7c, explain what "up, down, increase, decrease" mean in the figure legend.
- Not clear how this study suggests GC use in cancer therapy (Discussion)

Reviewers' comments:

Reviewer #1 (Remarks to the Author):

In this study, integrated GR ChIP-seq, H3K27ac ChIP-seq and HiChIP studies were performed in human podocytes. Comparisons are made between the podocyte data and published GR ChIP-seq data sets in two other cell types. ZBTB16 is defined as a podocyte-specific GR target and is implicated as a downstream transcriptional effector of cytoskeletal reorganization in response to induction of GR signaling.

Although there are numerous reports on GR ChIP-seq and other aspects of GR signaling in other cell types, there are no genome-wide reports on GR signaling in podocytes. Thus, the data presented here, based on what appear to be well-performed studies, will be a useful contribution to the literature. The conclusions made in the work are generally valid, although a more scholarly review of the literature encompassing additional references and current models of GR action would help contextualize these studies. Indeed, the two comparator cell types used in the primary analysis are not clearly representative of GR signaling responses in many other highly studied cell types, in which GR binding sites based on the consensus sequence strongly overlap with GR binding peaks defined by ChIP. In that regard, although ZBTB16 was not identified as a GR target in the two comparator cell types, it is well known as a target in other non-podocyte contexts. Thus, specific suggestions to improve the impact of the work include:

1) Please compare and contrast GR regulation in this cell type with several other GR ChIP-seq data sets that are potentially more representative, such as/including A549, HeLa and BEAS-2B cells. Please specifically include analysis of ZBTB16 in this work. In that regard, it would also be useful to determine whether GR interacts with the synpo locus, which would imply a GR-ZBTB16 feed forward loop.

Thanks for the suggestion. We now compare the GR binding in podocyte to that in additional cell type including A549, HeLa and BEAS-2B (Fig. 1a and Fig. S3). There are both shared and unique binding sites between podocyte and any of the other cell types as the original comparison between podocyte to MCF7 and K562.

Fig. 1

Fig. S3

Motif analysis reveal that GRE is still the most enriched motif at GR binding sites in podocyte (updated Fig. 1c).

The binding of GR at ZBTB16 locus is present in all the examined cell types (Fig. S9b), but there is no GR binding at SYNPO in any of these cell types (Fig. S9c). The binding of GR at ZBTB16 is not podocyte-specific, but the podocyte-specific enrichment of GRE-associated GR binding is one of the reasons enable us to discover the enrichment of responsive GR sites at super enhancers.

2) “Super enhancers” are cell type specific. The criteria that was used to define super enhancers in this work are not clear in the text (there are just three sentences in the supp methods), nor is it clear that dex-induced H3K27ac levels should be used as a criteria for super enhancer. Please provide more details in the text on this approach and justify why using dex-induced H3K27ac levels can be used to define a super enhancer – this seems like it could be a tautology.

Sorry for the typo. We called super enhancers based on H3K27ac ChIP-seq with or without DEX treatment independently. We then take the union of the super enhancer regions and examined the change of activities of all the potential super enhancers in Fig. 4c. The same procedure is carried out in A549 (Fig. 4f). We have revised the description in method ‘Super enhancer identification’ (supplementary line 100-line 106) and provide the super enhancer lists in Table S1.

3) Whether changes in three-dimensional interactions as assayed by HiC or similar approaches represent 1) changes in the efficiency of cross-linking of loops due to the absence or presence of transcription factors or 2) whether they in fact represent dynamic changes in chromatin structure, is not really clear. Please consider this first alternative in discussing induced and “repressed loops”

Thanks for the comments. We agree that HiChIP results reflect the combination effects of dynamic chromatin structure as well as the binding efficiency of a specific factor at even static chromatin structure. We now describe about the dynamics in chromatin interactions as in line 198- line 201 ‘The changes of three-dimensional interactions are a combinatory result with changes of H3K27ac signal at the anchors, as well as true physical interaction dynamics. Either way, it would affect the regulatory potential.’

4) In many places the language is difficult to follow and uses arcane terminology and sentence structure. Editorial review of the text is suggested.

We now revise the manuscript with help of a native English speaker.

5) For maximal utility to the community, data needs to be also mapped in hg38

Thanks for the suggestion. We now also map all the data in hg38 and all the map in hg38 are updated in GSE117888.

Reviewer #2 (Remarks to the Author):

In this manuscript, Wang et al use several genome-wide approaches to investigate GR action in a podocyte cell line model. The authors conclude that GR binds directly to many tissue-specific enhancers and that the GR cistrome is particularly enriched at super-enhancers. The authors also validate some of their findings by confirming the functional role of some novel target genes. While the work uses several interesting genomic approaches, I found some concerns regarding the experimental design chosen by the authors which need to be either addressed or properly justified as they may put into question some of their claims.

Major comments

1- The experimental design. The authors chose to work with a conditional transformed cell line model, wherein culturing these cells under 37C leads to podocyte maturation. According to ref.11, the maturation process takes 12-13 days without dexamethasone, but interestingly the differentiation process can be accelerated by DEX. In this manuscript, the authors only cultured the cells for 4 days, and then added DEX as a treatment for an additional 5 days. May I ask why the authors chose this protocol? My specific concerns are 1) the maturation process was not completed, therefore the authors were not working with podocytes before DEX treatment, and 2) DEX was likely involved in the maturation process, therefore the authors are not studying the effect of GCs on matured podocytes but rather the effect of DEX on podocyte differentiation. Have the authors found mature podocyte markers before adding DEX, such as synaptopodin or nephrin (Saleem et al JASN 2002)? In this regard, Fig. S1 shows a big difference in phenotype before and after DEX. If the authors follow the 10-14 day maturation protocol as in ref. 12 (no DEX at all), How the cells would compare in phenotype?

Thanks for the comments. As it is described in original ref. 11 and other references¹⁻³, the cells demonstrate a cobblestone appearance at 33°C. After they are mature under 37°C, they will turn to enlarged cuboidal, polygonal or multipolar shapes. We monitored the appearance of the cells during the culture. As it is showed in Fig. S1, the cells changed into enlarged irregular shapes after 4 days at 37°C (Fig. S1a).

The cells transcribed markers as synaptopodin and CD2AP at day 4 (Fig. S1b).

Thus, we added DEX at day 5 to minimize the effect of GC on maturation. However, it is still possible that GC further improve the maturation during the rest of the culture. During the rest of the culture, the shape of the cells were similar in culture with or without DEX, but the cells with DEX showed stronger cytoskeleton (Fig. S1a) and adhered to plate stronger. For both cells with or without DEX, the state turns bad after day 10. For the above observations, we choose current protocol.

2- The long DEX treatment. Performing a 5-day DEX treatment and then analyze genomic binding, epigenetic marks, and transcription can be tricky, as secondary effects are basically unavoidable (as the authors showed with the ZBTB16 and SYNPO genes). May I ask for the rationale for the longer treatment? What is the cell cycling duration on these cells? Do they grow during the 5-day treatment? I find hard to interpret the genomic data and transcriptional activity as any observed effect is likely due to secondary and even tertiary effects.

Thanks for the comments. The cells stopped proliferation at 37°C², and DEX did not increase the proliferation (Fig. S1c).

We choose long DEX treatment as we observed phenotype change (enhanced cytoskeleton) with this treatment, but not with 1h treatment. It is probably that GR is in a hit-and-run model to associate with chromatin. The binding of GR on chromatin would be lost when we change the cells to fresh medium without DEX during differentiation culture. To investigate whether the binding of GR in podocyte is primary or secondary, we now carry out ChIP-seq of GR after 1h treatment of DEX. The result showed that the binding profiles of GR were quite similar between 1h and 5d treatment (following figure a). The binding of GR at ZBTB16 was already there after 1h treatment (following figure b). The binding of GR would not be secondary effect. In contrast, there was no GR binding at SYNPO after 1h treatment, the same as the result after 5d treatment (following figure c). The regulation on genes not targeted by GR as SYNPO is secondary effect in both short or long-term treatment. In non-dividing podocyte, GR tended to bind at the GREs. It is previously reported the GR binding on chromatin lasting from short-term to long-term are those binding at GREs⁴. That would be the reason why short or long treatment did not affect much on the

GRE directed GR binding in podocyte.

3- The experimental replicates. There is only one GR ChIP replicate which potentially puts these findings on somewhat shaky grounds. Even though the authors stated that the GR profile is consistent with another hPC study (ref. 12), the experimental design is very different (i.e. hPC matured for 10-14 days before treatments, use of Prednisolone 1 h instead of Dex for 5 days). The same replicate issue applies to the H3k27ac ChIPs (only one replicate).

Thanks for the suggestion. We now performed replications for both GR and H3K27ac ChIP-seq. The results showed to be highly reproducible (Fig. S2a and Fig. S4a). GR displayed the same distribution in genome-wide and at genes as ZBTB16, SAA1 or SERPINE1 (Fig. S2c and Fig. S9b). The H3K27ac dynamics kept the same pattern in the replications (Fig. S4b). We updated all replications in GSE117888.

Fig. S2

Fig. S4

4- ChIP comparisons. The authors compared their GR ChIP dataset with previously published data in hPC and other cell lines. Could the relatively lower number of peaks detected (~1000 peaks) compared to the other cell lines be related to the very long DEX treatment? Once again, I find the comparisons difficult as the experimental designs are so different.

Thanks for the comments. We do not think the lower number of peaks in podocyte is due the long DEX treatment for the following observations. The GC treatment in McCaffrey study is 5h, and we performed GR ChIP-seq with 5d as well as 1h treatment. For all these treatments including both short and long treatments, the numbers of peaks in podocyte are low as around 1000. We think the lower number probably due to that podocytes stop proliferation after start of differentiation. We uploaded all the GR ChIP-seq at 1h and 5d in GSE117888.

5- Fig. 2. How comparable is the adult-kidney epigenome to mature podocytes? I am wondering how valid is to use the kidney datasets to segment the genome for the podocytes.

Thanks for the comment. We agree that whole kidney epigenome is not accurately comparable to podocyte epigenome. We now annotate the GR sites based on genomic feature as TSS (Fig. 2a).

Fig. 2

6- It is generally accepted that GR does not massively change the genome topology

(Ref. 9). I wonder whether the effects seen in the HiChIP H3K27 experiment are due to the experimental design chosen by the authors. In other words, would the authors have observed similar changes with a shorter DEX treatment, or the changes observed are a consequence of the differentiation process during the 5-day DEX treatment?

Among 41947 loops, we observed 260 strengthened and 287 weakened loops, accounting about 1% of all the loops. The chromatin topology was not massively changed in our study as that in Ref.9. Although the changed loops are a small fraction of all the loops, they cast transcriptional regulation roles.

HiChIP with H3K27ac antibody reflected the combination effects of dynamic chromatin structure as well as the binding efficiency of H3K27ac at even static chromatin structure. We found the GR sites in podocytes were enriched for GREs and worked as responsive enhancers to induce H3K27ac signal. As it is presented in answer to comment 2, GR associated with chromatin in a similar pattern with 1h treatment as that with 5d DEX treatment. We do not think the effect is due to long-term treatment.

Minor comments

1- Although the paper is readable, I strongly suggest the authors edit the manuscript with a native English speaker, as many grammar mistakes are present throughout the text.

Thanks for the suggestion. We now revise current version with language service from a native English speaker.

2- Fig. S2. Please add the total number of peaks shown in the heat maps. Also, in the legend and the figure, McCaffey should be McCaffrey.

Thanks for the notification. We now add the number of peaks as suggested and correct the typo. Sorry for that.

3- Fig. 1c and text. The authors compared motif enrichment between their dataset and others and concluded that "GR likely binds directly to GRE, rather than being indirectly recruited by other factors"; While I agree with the conclusion, the last paragraph in page 4 seems to imply that the presence of other motifs demonstrates indirect binding. This is not necessarily the case as the pioneer model (ref. 17) states that pioneer factors open the chromatin to allow the secondary factor (in this case GR) to bind directly to their response element. Perhaps the authors should rephrase that paragraph.

Thanks for the suggestion. We now rephrase the description about this element in line 136 as 'GR likely binds directly to GRE rather than being indirectly recruited by other factors or binding following other pioneer factors.'

4- Fig. S4 is not very informative as it stands. I suggest the authors invest more space to explain in more detail the chromatin segmentation technique.

'ChromHMM is a Java program for the learning and analysis chromatin states using a multivariate Hidden Markov Model that explicitly models the observed combination of marks' ⁵. As the reviewer suggested in major comments 5, we agree the whole kidney chromatin segmentation could not be exactly aligned to podocyte chromatin. We delete the chromatin segmentation. Instead, we annotate the GR site in podocyte based on genomic features (Fig. 2a), and demonstrate that a lot of GR sites are in non-promoter regions.

Reviewer #3 (Remarks to the Author):

1- The need to study GR specifically in podocytes is not properly discussed in the Introduction

Thanks for the notification. We now revise the introduction to present our motivation to study GR in podocyte in line 45- line 58. 'Podocyte, with exquisite actin cytoskeleton to maintain its structure, is critical for normal kidney function. It adheres to the outer surface of the glomerulus to form a normal glomerular filtration barrier and is sensitive to various stimulations. Any damage to its cytoskeleton will lead to disrupted filtration barrier, massive leakage of protein into urine, and progression of kidney disease ⁶. It is previously assumed that GCs act primarily by dampening the immune response to cure podocytopathy. However, increasing body of evidences indicate that GCs could attenuate podocytopathy by direct effect on podocyte ^{7,8}, by enhancing actin cytoskeleton ⁹. DEX treatment can recover disrupted podocyte cytoskeleton caused by LPS. However, with podocyte-specific deletion of GR, DEX cannot rescue the phenotype ¹⁰. Additionally, GR in other kidney epithelial cells like parietal epithelial cells is not required for podocyte homeostasis ¹¹. Therefore, GR in podocyte mediates a direct protective effect by enhancing cytoskeleton. However, it is understudied how GR accomplishes this task.'

2-Define "responsive GR site" versus "non-responsive GR site"

The 'responsive GR site' is defined by previous study¹². 'The responsive sites were defined by direct GR binding via a GC response element (GRE) and exclusively increased reporter-gene expression', 'H3K27ac and H3K4me1 increased significantly more in the flanks of the DEX-responsive GBSs than in flanks of the non-responsive GBSs', and 'direct GR binding to a GRE is predictive of DEX-inducible reporter activity'. In this study, we found the GR sites in podocytes enriched for GRE (Fig. 1c). Additionally, we observed prominent increase of H3K27ac at GR sites after DEX treatment (Fig. 2c). Thus, we concluded that GR sites in hPC display features as responsive GR sites.

3-Not clear why the authors are comparing hPC treated with DEX vs GC prednisolone (Fig. S2)

As both DEX and prednisolone are GC, we checked against the distribution of GR with prednisolone as quality control. As we see from the result, the signals are consistent in podocyte. The ChIP-seq is reproducible, and the data from two independent lab remains consistent (Fig. S2).

4- Why are authors looking at BIN1 and IRS2 in Fig S6B?

These are two representative examples for the scenarios 1) genes without GR binding nearby, but interact with a distant GR site, and the expression is upregulated; 2) genes with GR site in the gene body, but the GR site did not interact with its promoter, and the gene is not upregulated. We further examined the two scenarios in Fig. 3c. We now revise the figure legend to explain it.

5-Define anchor (Fig. 3a). Explain why there is no difference +/-DEX in Fig 3a+3b

An Anchor is one end of the chromatin interaction. 'Each chromatin interaction detected by an inter-ligation PET cluster features two anchor regions (interacting loci)¹³. According to the principle of HiChIP experiment¹⁴, the anchors should enrich for the specific protein targeted by the antibody (H3K27ac in this study).

With H3K27ac antibody, we intended to enrich the interactions between enhancers or promoters. We are glad that the experiments work well, and we successfully capture these features. Fig. 3a confirmed the anchor of interactions in this study did enrich for H3K27ac, and Fig. 3b showed the interactions enriched for those between enhancers and promoters. The protocol is highly reproducible, as there is not genome-wide shift on different experiments as with or without DEX treatment.

6-In Fig.S7c, explain what "up, down, increase, decrease" mean in the figure legend.

Thanks for the suggestion. By ‘increase’ and ‘decrease’, we described the change of interactions. Differential interactions were calculated with edgeR. ‘Increase’ loops were those with $pvalue < 0.05$ and $\log(\text{foldchange}) > 0$, and ‘decrease’ loops were those with $pvalue < 0.05$ and $\log(\text{foldchange}) < 0$. To prevent confusion, we now use ‘strengthened’ or ‘weakened’ to describe the interactions.

By ‘up’ and ‘down’, we described the change of gene expression. Differential expression was calculated with DESeq2. ‘Up’ expressed genes were those with $\text{adjust } p < 0.01$ and $\log(\text{foldchange}) > 0$, and ‘down’ expressed genes were those with $\text{adjust } p < 0.01$ and $\log(\text{foldchange}) < 0$. We have revised the figure legend to better explain the figure.

7-Not clear how this study suggests GC use in cancer therapy (Discussion)

Our data suggest that GR binding on chromatin could enrich at super enhancers. Cancer cells are often addicted to the super-enhancer driven transcriptional programs. Pharmacological targeting against super-enhancers has shown great promise in reducing tumor growth and proliferation in several pre-clinical tumor models.^{15,16} We think the use of GC might rewire the super enhancer profile in cancers and create a genomic environment facilitating therapies. In lines with this hypothesis, it has been demonstrated that add GC into the therapy could improve ER+ Breast Cancer Outcome¹⁷. We revised this part in discussion line 326- line 332 to better explain our thinking. Thanks.

- 1 Xing, C. Y. *et al.* Direct effects of dexamethasone on human podocytes. *Kidney international* **70**, 1038-1045, doi:<https://doi.org/10.1038/sj.ki.5001655> (2006).
- 2 Saleem, M. A. *et al.* A Conditionally Immortalized Human Podocyte Cell Line Demonstrating Nephritin and Podocin Expression. *Journal of the American Society of Nephrology* **13**, 630-638 (2002).
- 3 Sakairi, T. *et al.* Conditionally immortalized human podocyte cell lines established from urine. *American Journal of Physiology-Renal Physiology* **298**, F557-F567, doi:10.1152/ajprenal.00509.2009 (2010).
- 4 McDowell, I. C. *et al.* Glucocorticoid receptor recruits to enhancers and drives activation by motif-directed binding. *Genome research*, doi:10.1101/gr.233346.117 (2018).
- 5 Ernst, J. & Kellis, M. ChromHMM: automating chromatin-state discovery and characterization. *Nature methods* **9**, 215-216, doi:10.1038/nmeth.1906 (2012).
- 6 He, S. K. M. J. C. The podocyte as a direct target for treatment of glomerular disease? *Am J Physiol Renal Physiol* **311**, F46-F51, doi:10.1152/ajprenal.00184.2016.-The (2016).
- 7 Xing, C. Y. *et al.* Direct effects of dexamethasone on human podocytes. *Kidney international* **70**, 1038-1045, doi:10.1038/sj.ki.5001655 (2006).
- 8 Schonenberger, E., Ehrlich, J. H., Haller, H. & Schiffer, M. The podocyte as a direct target of immunosuppressive agents. *Nephrology, dialysis, transplantation : official publication of the European Dialysis and Transplant Association - European Renal Association* **26**, 18-24, doi:10.1093/ndt/gfq617 (2011).
- 9 Ransom, R. F., Lam, N. G., Hallett, M. A., Atkinson, S. J. & Smoyer, W. E. Glucocorticoids protect and enhance recovery of cultured murine podocytes via actin filament stabilization. *Kidney international* **68**, 2473-2483, doi:10.1111/j.1523-1755.2005.00723.x (2005).
- 10 Zhou, H. *et al.* Loss of the podocyte glucocorticoid receptor exacerbates proteinuria after injury. *Scientific reports* **7**, 9833, doi:10.1038/s41598-017-10490-z (2017).
- 11 Kuppe, C. *et al.* Investigations of Glucocorticoid Action in GN. *Journal of the American Society of Nephrology : JASN* **28**, 1408-1420, doi:10.1681/ASN.2016010060 (2017).
- 12 Vockley, C. M. *et al.* Direct GR Binding Sites Potentiate Clusters of TF Binding across the Human Genome. *Cell* **166**, 1269-1281.e1219, doi:<https://doi.org/10.1016/j.cell.2016.07.049> (2016).
- 13 Fullwood, M. J. *et al.* An oestrogen-receptor- α -bound human chromatin interactome. *Nature* **462**, 58-64, doi:10.1038/nature08497 (2009).
- 14 Mumbach, M. R. *et al.* HiChIP: efficient and sensitive analysis of protein-directed genome architecture. *Nature Methods* **13**, 919-922, doi:10.1038/nmeth.3999 (2016).
- 15 Lovén, J. *et al.* Selective Inhibition of Tumor Oncogenes by Disruption of Super-Enhancers. *Cell* **153**, 320-334, doi:<https://doi.org/10.1016/j.cell.2013.03.036> (2013).

- 16 Bradner, J. E., Hnisz, D. & Young, R. A. Transcriptional Addiction in Cancer. *Cell* **168**, 629-643, doi:<https://doi.org/10.1016/j.cell.2016.12.013> (2017).
- 17 West, D. C. *et al.* GR and ER Coactivation Alters the Expression of Differentiation Genes and Associates with Improved ER+ Breast Cancer Outcome. *Molecular cancer research : MCR* **14**, 707-719, doi:10.1158/1541-7786.MCR-15-0433 (2016).

Reviewers' comments:

Reviewer #1 (Remarks to the Author):

This manuscript focuses on GR signaling in podocytes using ChIP-seq and other methods. The revised manuscript has novel areas and is improved. It addresses some of the previous concerns, however, there remain several areas that dampen potential impact of the work. First, although the language is improved, there are still some instances of non-standard English. Additional editing for grammar and readability would be useful.

With respect to scientific issues, 1. the reported unusually low number of sites of GR occupancy in this cell type; 2. the very high relative representation of high affinity GREs within the set of GR-occupied sites, and 3. the purported greater than expected cell type specificity, is of high interest both to podocyte biology and also to the greater GR research community. Unfortunately, the current comparisons (some of which were made in response to initial critiques regarding contentions about the unique podocyte GR program) create yet further questions about the validity of these observations. For example, my review of the HeLa cell dataset indicates the presence of a clear, MACS2-defined steroid-induced GR binding site in the SAA1 locus. It is certainly less intense than the sites near PER1 in HeLa cells, but it is present and would be included as an occupancy site in standard ChIP-seq analysis. If, clear sites of GR occupancy akin to the HeLa SAA1 site are being excluded in the reported analysis of GR occupancy in podocytes, the analysis needs to be updated to reflect more standard peak calls and overlap analysis with other datasets. Similarly, review of the BEAS-2B datasets reveals that the authors have used a 10-minute steroid-exposure ChIP-seq dataset for comparison rather than also published one hour data. Not surprisingly, the one hour data shows significantly more GR peaks and is clearly a better comparison. Indeed, the one-hour BEAS-2B data has a very strong site of GR occupancy at SAA1. Likewise, the ENCODE annotation for GR occupancy in A549 cells at the same SAA1 site also indicates a moderately strong site of GR occupancy. Moreover, for SNAI2, just outside the area of the screenshot, there are two strong GR peaks in the HeLa cell dataset. Based on these comparisons/discrepancies, the manuscript could have a higher impact if the authors 1. review their peak calling in detail and 2. make certain that the reported high cell type specificity and relatively low peak number (with a very high representation of strong GREs) in podocytes is based on appropriate and fair statistical comparisons with other published datasets. The issue is not what the final data show with respect to peak number or cell type specificity in this cell type; the issue is making sure that the comparisons are accurate.

Reviewer #2 (Remarks to the Author):

The authors have addressed all my concerns.

Please review some minor issues with symbols (e.g. uM instead of the correct μM , line 85 of the marked version).

Finally, while some grammar issues have been corrected, the main text still needs more refinement.

Reviewers' comments:

Reviewer #1 (Remarks to the Author):

This manuscript focuses on GR signaling in podocytes using ChIP-seq and other methods. The revised manuscript has novel areas and is improved. It addresses some of the previous concerns, however, there remain several areas that dampen potential impact of the work. First, although the language is improved, there are still some instances of non-standard English. Additional editing for grammar and readability would be useful.

Thanks for the suggestion. We revise the manuscript with language service from a native English speaker.

With respect to scientific issues, 1. the reported unusually low number of sites of GR occupancy in this cell type; 2. the very high relative representation of high affinity GREs within the set of GR-occupied sites, and 3. the purported greater than expected cell type specificity, is of high interest both to podocyte biology and also to the greater GR research community. Unfortunately, the current comparisons (some of which were made in response to initial critiques regarding contentions about the unique podocyte GR program) create yet further questions about the validity of these observations. For example, my review of the Hela cell dataset indicates the presence of a clear, MACS2-defined steroid-induced GR binding site in the SAA1 locus. It is certainly less intense than the sites near PER1 in Hela cells, but it is present and would be included as an occupancy site in standard ChIP-seq analysis. If, clear sites of GR occupancy akin to the Hela SAA1 site are being excluded in the reported analysis of GR occupancy in podocytes, the analysis needs to be updated to reflect more standard peak calls and overlap analysis with other datasets. Similarly, review of the BEAS-2B datasets reveals that the authors have used a 10-minute steroid-exposure ChIP-seq dataset for comparison rather than also published one hour data. Not surprisingly, the one hour data shows significantly more GR peaks and is clearly a better comparison. Indeed, the one-hour BEAS-2B data has a very strong site of GR occupancy at SAA1. Likewise, the ENCODE annotation for GR occupancy in A549 cells at the same SAA1 site also indicates a moderately strong site of GR occupancy. Moreover, for SNAI2, just outside the area of the screenshot, there are two strong GR peaks in the Hela cell dataset. Based on these comparisons/discrepancies, the manuscript could have a higher impact if the authors 1. review their peak calling in detail and 2. make certain that the reported high cell type specificity and relatively low peak number (with a very high representation of strong GREs) in podocytes is based on appropriate and fair statistical comparisons with other published datasets. The issue is not what the final data show with respect to peak number or cell type specificity in this cell type; the issue is making sure that the comparisons are accurate.

Thanks for the comments. As reviewer suggested, we now use GR ChIP-seq in BEAS-2B from the one with 1h treatment (SRR8485261) instead of the one with 10min treatment. The GR ChIP-seq in other cell types include MCF7 (SRR2176969), K562

(E-MTAB-2955), A549 (SRR5093186) and HeLa (SRR067992). We used standard method as MACS2 for peak calling of GR in podocyte as well as that in other cell types. We used default parameters except -q. In previous version, we used -q as 0.05 for GR in other cell type and -q as 0.01 for GR in podocyte, to guarantee that the unshared podocyte GR sites are high confident peaks. In this version, we used -q as 0.01 for all GR ChIP-seq to repeat the analysis. We add analysis parameters in ‘Methods: Bioinformatics: ChIP-seq analysis’. Although the numbers of analysis changed in a small range, all the conclusions remain true. We will explain the results further in following sections.

We don’t think the lower number of GR sites in podocyte is due to peak calling statistics. The number of GR peak is around 1000 with -q as either 0.01 or 0.05. Additionally, the number of GR sites in an independent study carried out by McCaffrey et al. is also around 1000. We think the lower number probably due to that podocytes, unlike other cell types, do not proliferate during study.

The comparisons between podocyte and other cell types in Figure 1b and Supplementary Figure 3 were carried out by pairwise overlapping of peaks. The results are presented in Supplementary Data 1. The proportion of GR peaks that are shared by the other cell type range from 16.9 to 78.6% (19.6% to 57.8% in previous version). The level of peak overlapping varies among cell types. There are cell types like MCF7 that are quite different from podocyte, as 83.1% of hPC GR sites are not present in MCF7. There are also cell types like BEAS-2B showing similarity of GR binding to that in podocyte, as 21.4% of hPC GR sites are not present in BEAS-2B.

The motifs enrichment of each group of GR sites in Figure 1c were performed with HOMER with parameters ‘-size given’. The groups of GR sites were identified among all the six cell types instead of pairwise comparison as that in Figure 1b. The groups of peaks are listed in Supplementary Data 2. GR motif is still enriched in common and specific podocyte GR sites in current analysis. It is previously reported the GR binding lasting from short-term to long-term are tended to enriched for GREs¹. The discovery of GRE on GR site in podocyte is consistent with previous report as GR sites in non-dividing podocyte in this study are long-term binding sites.

In previous Figure 1a, we showed *SAA1* and *SNAI2* loci as examples for non-sharing GR sites, we did not use peak in *SAA1* locus as podocyte-specific peak in Figure 1c (the specific peaks are listed in Supplementary Data 2). As review suggested and present in following figures, there are weak peaks in A549 and HeLa, and strong peaks in BEAS-2B (1h), but no peak in MCF7 and K562. Although *SAA1* is a non-shared GR site, we replace it with cell type-specific sites to better represent the non-sharing GR sites. For *SNAI2* locus, we could observe two peaks in HeLa, but the peaks in either MCF7, HeLa or BEAS-2B are cell type-specific peaks.

We replaced *SAA1* and *SNAI2* with another two example loci to show non-sharing GR sites in Figure 1a. As most of the podocyte-specific GR sites located in intergenic regions, we present an intergenic peak for podocyte-specific GR sites. We also show MCF7-specific GR binding at *PKIA* locus as example of non-sharing GR binding.

Reviewer #2 (Remarks to the Author):

The authors have addressed all my concerns.

Please review some minor issues with symbols (e.g. uM instead of the correct μM , line 85 of the marked version).

Thanks for remind. We have corrected the symbols.

Finally, while some grammar issues have been corrected, the main text still needs more refinement.

Thanks for the suggestion. We revise the manuscript with language service from a native English speaker.

- 1 McDowell, I. C. *et al.* Glucocorticoid receptor recruits to enhancers and drives activation by motif-directed binding. *Genome research*, doi:10.1101/gr.233346.117 (2018).

REVIEWERS' COMMENTS:

Reviewer #1 (Remarks to the Author):

All of my concerns are addressed, and I now find it easier to contextualize these data into other studies of GR signaling.